# Neuroprotection of Andrographolide against Neurotoxin MPP^+^-Induced Apoptosis in SH-SY5Y Cells via Activating Mitophagy, Autophagy, and Antioxidant Activities

**DOI:** 10.3390/ijms24108528

**Published:** 2023-05-10

**Authors:** Prachayaporn Prasertsuksri, Pichnaree Kraokaew, Kanta Pranweerapaiboon, Prasert Sobhon, Kulathida Chaithirayanon

**Affiliations:** 1Department of Anatomy, Faculty of Science, Mahidol University, Bangkok 10400, Thailand; prachayaporn.prs@student.mahidol.ac.th (P.P.);; 2Chulabhorn International College of Medicine, Thammasat University, Pathumthani 12120, Thailand

**Keywords:** Parkinson’s disease, andrographolide, MPP^+^, SH-SY5Y cells, ROS, Nrf2, mitophagy

## Abstract

Parkinson’s disease (PD) is associated with dopaminergic neuron loss and alpha-synuclein aggregation caused by ROS overproduction, leading to mitochondrial dysfunction and autophagy impairment. Recently, andrographolide (Andro) has been extensively studied for various pharmacological properties, such as anti-diabetic, anti-cancer, anti-inflammatory, and anti-atherosclerosis. However, its potential neuroprotective effects on neurotoxin MPP^+^-induced SH-SY5Y cells, a cellular PD model, remain uninvestigated. In this study, we hypothesized that Andro has neuroprotective effects against MPP^+^-induced apoptosis, which may be mediated through the clearance of dysfunctional mitochondria by mitophagy and ROS by antioxidant activities. Herein, Andro pretreatment could attenuate MPP^+^-induced neuronal cell death that was reflected by reducing mitochondrial membrane potential (MMP) depolarization, alpha-synuclein, and pro-apoptotic proteins expressions. Concomitantly, Andro attenuated MPP^+^-induced oxidative stress through mitophagy, as indicated by increasing colocalization of MitoTracker Red with LC3, upregulations of the PINK1–Parkin pathway, and autophagy-related proteins. On the contrary, Andro-activated autophagy was compromised when pretreated with 3-MA. Furthermore, Andro activated the Nrf2/KEAP1 pathway, leading to increasing genes encoding antioxidant enzymes and activities. This study elucidated that Andro exhibited significant neuroprotective effects against MPP^+^-induced SH-SY5Y cell death in vitro by enhancing mitophagy and clearance of alpha-synuclein through autophagy, as well as increasing antioxidant capacity. Our results provide evidence that Andro could be considered a potential supplement for PD prevention.

## 1. Introduction

Parkinson’s disease (PD) is a progressive neurological disorder frequently found in elderlies. Globally, the number of individuals with PD has more than doubled to over 6 million. This disease mainly affects motor and non-motor systems due to the loss of dopaminergic neurons (DA) in the substantia nigra pars compacta and striatal terminal, respectively [1]. The loss of DA is caused by aging, gene mutations, and neurotoxin exposures such as 1-methyl-4-phenylpyridinium (MPP^+^), 6-hydroxydopamine (6-OHDA), and agricultural toxins. In particular, neurotoxin MPP^+^ has been widely applied in the in vitro cellular models of PD owing to its ability to generate excessive reactive oxygen species (ROS) [2]. The ensuing oxidative stress causes depolarization of mitochondrial membrane potential (∆ψM), which leads to the impairment of mitochondria through the inhibition of the electron transport chain (ETC) at the complex I, resulting in ATP depletion and thus initiating the death of DA neurons [3,4,5,6]. In addition, a recent study has shown that MPP^+^ increases the accumulation of misfolded alpha-synuclein, which contributes further to PD progression [7].

Normally, mitophagy helps regulate mitochondrial number and quality by eliminating damaged or dysfunctional mitochondria by using common mediators with the autophagic process [8]. In addition, PTEN-induced kinase 1 (PINK1) and Parkin RBR E3 ubiquitin-protein ligase (Parkin) are markers of dysfunctional mitochondria destined for mitophagy in neurodegenerative diseases such as PD [9,10,11]. In this regard, the dimerization of PINK1 at the outer mitochondrial membrane (OMM) recruits and phosphorylates Parkin, followed by ubiquitination (Ub) of damaged mitochondria. Poly-Ub chains are subsequently phosphorylated by PINK1 and serve as an ‘eat me’ signal for the autophagic machinery. Adaptor proteins (p62, OPTN (optineurin), NDP52 (nuclear dot protein 52 kDa) recognize phosphorylated poly-Ub chains, resulting in the initiation of autophagosome around dysfunctional mitochondria and binding with LC3, which leads to fusion with lysosome and degradation. Thus, autophagy is an important mechanism that plays a major role in removing damaged organelles (such as mitochondria) and proteins (such as misfolded alpha-synuclein) via autophagosome–lysosome fusion and degradation. Impaired autophagy leads to accumulations of dysfunctional mitochondria and alpha-synuclein, which subsequently leads to neuron death.

Various medications such as levodopa, carbidopa, bromocriptine, entacapone, and deep brain stimulation have been applied to cure PD. However, these pharmacologic agents provide only symptomatic relief of PD, but not cure [12,13]. Hence, discoveries of natural products may prevent or decelerate the progression of PD. The potential anti-PD action of natural products could be conveniently screened by determining their ability to attenuate damage in a cellular model of PD, such as MPP^+^-induced SH-SY5Y cells. Interestingly, andrographolide (Andro), a natural product isolated from *Andrographis paniculata*, was found to exhibit many pharmacological activities, especially antioxidant and anti-inflammatory properties [14]. For instance, Andro could increase the expression of nuclear factor erythroid 2-related factor 2 (Nrf2)/heme oxygenase 1 (HO-1) in rats with middle cerebral artery occlusion (MCAO)-induced ischemic stroke, which upregulated p38 MAPK signaling [15]. Mechanistically, dissociation of the KEAP1/Nrf2 complex induces Nrf2 nuclear translocation, which triggers the transcription of antioxidant enzymes such as heme oxygenase-1 (HO-1), superoxide dismutase (SOD), catalase (CAT), and glutathione peroxidase (GPx) [16]. Moreover, Andro ameliorated the inflammatory response in microglia by activating the Nrf2/HO-1 pathway and inhibiting NF-κB expression [17]. More importantly, a recent study demonstrated that Andro suppresses NLRP3 inflammasome activation in MPP^+^ and LPS-induced microglia through the induction of Parkin-mediated mitophagy in in vitro and in vivo models of Parkinson’s disease [18]. Nevertheless, whether Andro could provide neuroprotection against MPP^+^ insult in human neuroblastoma SH-SY5Y cell line remains uninvestigated. Therefore, in the present study, we investigated whether Andro could prevent MPP^+^-induced death of SH-SY5Y cells, serving as an in vitro model of PD, via inductions of mitophagy–autophagy and antioxidant mechanisms.

## 2. Results

### 2.1. Andrographolide Mitigates Death of Human Neuroblastoma SH-SY5Y Cells Induced by MPP^+^

To study the neuroprotection of Andro, we first evaluated the cytotoxicity effect of 0.02% of DMSO, a solvent of Andro. There was no cytotoxic effect in that group in comparison with the control (Appendix A). Thus, DMSO at 0.02% was used as a control in these experimental studies. Various concentrations (0.1, 0.25, 0.5, 0.75, 1, 1.25, 1.5, 2, 2.5, and 3 µM) of Andro did not show any cytotoxicity towards the neuroblastoma cell line SH-SY5Y after 24 h when compared to the control group. Interestingly, a drastic increase in cell viability was observed at the concentration of 1.5 µM of Andro (Figure 1A). Conversely, Andro at 6 µM could contribute to cytotoxicity in these cells. 

Moreover, Andro has no cytotoxic activity on normal cell line, such as human foreskin fibroblast (HFF) cells (Figure 1B). To determine the effective concentration of Andro for neuroprotection, SH-SY5Y cells were pretreated with Andro at 0.5, 1, 1.25, 1.5, 2, and 2.5 µM for 24 h and then exposed to 1.5 mM MPP^+^ for 24 h. We found that 1.5 µM of Andro pretreatment significantly ameliorated MPP^+^-induced neuronal cell death as compared with the MPP^+^ exposure group (Figure 1C). Therefore, 1.5 µM of Andro was selected to investigate the neuroprotective effects of Andro in further studies.

### 2.2. Andrographolide Ameliorates MPP^+^-Induced SH-SY5Y Cells Apoptosis through Protecting Mitochondrial Dysfunction

To examine the protective effect of Andro on MPP^+^-induced SH-SY5Y cells, the mitochondrial membrane potential (MMP) depolarization was detected by JC-1 staining as an indicator of cell apoptosis. JC-1 forms aggregates in energized mitochondria, which exhibits red fluorescence (dimer) in healthy cells. In contrast, the apoptotic or unhealthy cells with low MMP retain the original green fluorescence (monomer) [19]. MPP^+^ is one of the neurotoxins that cause MMP depolarization, as detected in Parkinson’s disease [5,20,21,22,23]. Remarkably, the cells pretreated with Andro at 1.5 µM showed significantly increased red fluorescence intensity which indicates healthy mitochondria, while the cells treated with MPP^+^ at 1.5 mM alone showed a significantly reduced intensity of red fluorescence while increasing that of green fluorescence, indicating the occurrence of MMP depolarization (Figure 2A,B). To further explore the impact of Andro on the apoptosis pathway during the course of MPP^+^ induction, we observed the expressions of proteins in MPP^+^-induced SH-SY5Y cells. MPP^+^ exposure significantly increases not only the levels of pro-apoptotic proteins, including cleaved-caspase-3, cytochrome c, and Bax, but also a pathogenic hallmark of PD, alpha-synuclein. Notably, pretreatment with Andro significantly downregulated these proteins as reflected by the decreases in protein expression in cleaved-caspase-3, Bax, and alpha-synuclein in comparison to the MPP^+^-treated group. Moreover, pretreatment with Andro tended to decrease cytochrome c when compared with MPP^+^ at 1.5 mM alone. Additionally, pretreatment with Andro significantly increased Bcl2 expression compared with the untreated control and MPP^+^-treated groups (Figure 2C,D). In order to investigate the neuroprotective effect of Andro on MPP^+^-induced SH-SY5Y cells, the expression of tyrosine hydroxylase (TH), which is an enzyme that catalyzes dopamine synthesis, was also observed. We found that pretreatment with Andro significantly increased TH compared with the control and MPP^+^-treated groups (Figure 2C,D). Accordingly, these findings suggest that Andro ameliorated MPP^+^-induced apoptosis in SH-SY5Y cells by preventing mitochondrial dysfunction.

### 2.3. Andrographolide Enhanced Mitophagy and Autophagy Induction to Eliminate Damaged Mitochondrial and Accumulation of Alpha-Synuclein

Autophagy impairment leads to accumulations of damaged mitochondria and toxic alpha-synuclein [24,25,26]. After treatment with MPP^+^, the morphology of mitochondria was observed by staining with MitoTracker Red. We found that after treatment with 1.5 μM of Andro, mitochondria appeared as interconnected red filamentous structures similar to those of the control group. In contrast, cells treated with MPP^+^ exhibited abnormal mitochondria that appeared and fragmented condensed red dots (Appendix A). Therefore, we studied the mitophagic effect, a clearance of defective mitochondria, under MPP^+^-induced toxicity, including pretreatment with Andro. To demonstrate whether pretreatment with Andro stimulated mitophagy, the colocalization of mitochondria and autophagosomes in SH-SY5Y cells was performed by staining with MitoTracker Red (a marker for mitochondria) and LC3 (a marker for autophagosome). In order to analyze the colocalization of mitochondria and autophagosomes, Manders’ coefficient (MCC) was used to quantify the degree of colocalization between the two markers. This MCC is calculated by dividing the number of pixels in which the two structures colocalized, resulting in a value ranging from 0 to 1 that reflects the degree of colocalization [27]. It was found that pretreatment with Andro at 1.5 µM significantly increased the intensity of MCC compared with the MPP^+^-treated group (Figure 3A,B). Moreover, there was an increased LC3 puncta in the pretreatment with Andro as compared with the MPP^+^-treated group. In a previous report [18], Andro inhibited inflammation in microglia by promoting mitophagy via the activation of Parkin. To further assess how Andro triggers mitochondrial autophagy, 3-methyladenine (3-MA), an autophagy inhibitor, was used to block the initiation of autophagy [28,29]. The results revealed that the ratio of p-mTOR/mTOR protein expression, which is a negative indicator of autophagy, was decreased in pretreatment with Andro compared with the MPP^+^-treated group, whereas SH-SY5Y cells exposed to 3-MA followed by pretreatment with Andro showed a slightly increased p-mTOR/mTOR ratio when compared with Andro pretreatment. In autophagy, LC3-phosphatidylethanolamine conjugate (LC3), the mammalian ortholog of the yeast autophagy-related gene 6 (Beclin1), and lysosomal-associated membrane protein 1 (LAMP1) are involved in the autophagosome formation. We found that pretreatment with Andro tended to increase LC3B/A and LAMP1 and significantly increased Beclin1 in comparison with MPP^+^ exposure. Correspondingly, 3-MA significantly decreased the expression of LC3B/A compared to pretreatment with Andro. As reported previously [30], Andro could attenuate chronic unpredictable mild stress-induced depressive-like behavior in mice through upregulation of autophagy. To corroborate this finding, p62/SQSTM1, an autophagosome-linked cargo protein that signifies the fusion of autophagosomes with lysosomes, was measured by Western blot, which showed a significant increase in p62 in the MPP^+^-treated group compared with the control group. This increase in p62 indicates the impairment of autophagosome degradation by the accumulation of p62, whereas pretreatment with Andro decreased p62 expression, as compared with the MPP^+^-treated group.

We next investigated the protective effect of Andro on mitophagy induction during MPP^+^ treatment of SH-SY5Y cells. PINK1 and Parkin are important factors that regulate mitophagy [31]. Pretreatment with Andro significantly increased PINK1 expression and tended to increase the expression of Parkin when compared with the MPP^+^-treated group. In contrast, exposure with 3-MA followed by pretreatment with Andro compromised the effect of Andro in facilitating mitophagy induction by significantly decreasing expression of PINK1 and Parkin when compared with pretreatment with Andro without prior 3-MA treatment (Figure 3C,D). Thus, these results suggested that Andro has a neuroprotective effect by enhancing autophagy and mitophagy, which facilitates clearances of alpha-synuclein and damaged mitochondria.

### 2.4. Andrographolide Exerted Antioxidant Effect against ROS Generated in MPP^+^-Treated SH-SY5Y Cells via Nrf2 Activation

The amount of intracellular ROS generated as the result of MPP^+^ treatment was measured by live-cell imaging and a microplate reader. We found that intracellular ROS accumulation was visibly increased in the MPP^+^ exposure group compared with the control. In contrast, Andro pretreatment reduced the intensity of intracellular ROS as compared with the MPP^+^-treated group (Figure 4A). Correspondingly, the quantification of ROS measured by microplate reader was significantly reduced in Andro pretreated samples compared to the MPP^+^-treated group (Figure 4C). We next demonstrated the activation of transcription factor Nrf2, which plays a key role in the antioxidant pathway, through its nuclear translocation by examining the colocalization of Nrf2 and DAPI in the nuclei of SH-SY5Y cells. We found that MPP^+^ treatment significantly diminished the Nrf2 nuclear translocation as compared with the control group, whereas Andro pretreatment at 1.5 µM significantly increased Nrf2 nuclear translocation compared with the MPP^+^-treated group (Figure 4B,D). These findings implied that Andro reduced ROS generation by promoting nuclear translocation, leading to the activation of Nrf2.

### 2.5. Andrographolide Activated Nrf2, Which Leads to the Upregulations of Antioxidant Enzymes to Counteract Oxidative Stress in MPP^+^-Treated SH-SY5Y Cells

In order to investigate the effect of Andro on Nrf2 activation, antioxidant protein and gene expressions, including antioxidant activities, were determined. NAC is commonly used as an antioxidant inducer [32]; thus, NAC was used as a positive control in the experiments. When analyzed by Western blot, the expression of KEAP1 levels was significantly decreased in the cells pretreated with Andro compared with the control and MPP^+^-treated groups, while the Nrf2 nuclear translocation was increased in the Andro pretreatment group as compared with the MPP^+^-treated group. Accordingly, the expression of HO-1 levels was significantly increased in the cells pretreated with Andro compared with the control and MPP^+^-treated groups (Figure 5A,B), while treatment with NAC tended to increase Nrf2 nuclear translocation and HO-1 compared with the control group. Next, to verify whether Andro is able to activate the expression of genes encoding antioxidant enzymes activity during the course of MPP^+^ treatment, RT-qPCR was performed. We found that the MPP^+^-treated group downregulated mRNA expressions of SOD1 and GSTP1 compared with the control group. In contrast, pretreatment with Andro drastically upregulated SOD1, CAT, GSTP1, and HO-1 gene expression compared with the MPP^+^-treated group (Figure 5C), while treatment with NAC did not exhibit a significant difference compared with the control group. Thus, these results indicated that Andro induced antioxidant activity by upregulating antioxidant genes against ROS generated in MPP^+^-treated SH-SY5Y cells. Furthermore, the activities of SOD and GPx antioxidant enzymes were evaluated, as shown in Figure 5D, where the activities of SOD and GPx enzymes in Andro pretreatment were significantly upregulated in comparison with the MPP^+^-treated group. Thus, these data confirmed that in addition to activations of autophagy and mitophagy, Andro could also activate antioxidant activities via Nrf2/HO-1 signaling pathway to attenuate oxidative stress in MPP^+^-treated SH-SY5Y cells.

## 3. Discussion

Despite many studies reporting on various pharmacological activities of andrographolide, we are the first to report that Andro pretreatment could prevent MPP^+^-induced SH-SY5Y cell death by promoting mitophagy along with autophagy through its ability to activate PINK-Parkin and LC3 B/A, respectively. Moreover, Andro could decrease intracellular oxidative stress by activating Nrf2, which upregulated the expression of genes encoding antioxidant enzymes, thereby alleviating cell damage and death.

We have shown that MPP^+^ at 1.5 mM triggered mitochondrial damage in SH-SY5Y cells that led to increased ROS production, resulting in increased mitochondrial membrane depolarization, which caused cell apoptosis. These defective mitochondria were removed by the Andro-initiated mitophagy and autophagy as demonstrated by colocalization of MitoTracker red and LC3, as well as the upregulations of PINK1 and Parkin proteins in MPP^+^-treated SH-SY5Y cells that were pretreated with Andro. These results were similar to those reported previously by Ahmed et al. [18], where Andro prevented the loss of dopaminergic neurons in NLRP inflammasome-induced microglial-mediated neuroinflammation in PD by promoting mitophagy of defective mitochondria. Moreover, it has been reported that Andro protects neurons by inhibiting excessive dynamin-related protein 1 (DRP1)-mediated mitochondrial fission, which is responsible for maintaining the function of mitochondria against rotenone- or MPTP-induced damage in in vitro and in vivo models, respectively [33]. Furthermore, we also demonstrated that Andro decreased the expression of p-mTOR, which is a negative regulator of autophagy. Then, the autophagy-related proteins that are involved in autophagosome–lysosome formation were increased, including Beclin1, LC3 B/A, and LAMP1. Defective mitochondria expressed PINK1 on the outer membrane, followed by Parkin, which was subsequently ubiquitinylated and recruited autophagy-forming molecules such as p62 and LC3 B/A to initiate mitophagy. Meanwhile, our study revealed that MPP^+^ exposure slightly upregulated autophagy-related proteins but impaired autophagosome degradation, as shown by the accumulation of p62 in cells exposed to only MPP^+^. This accumulation of p62 leads to autophagy impairment which leads to the accumulation of mitochondria and protein dysfunction. In consonance with Sakamoto and Rokad [28,34], human neuroblastoma SH-SY5Y cells were exposed to low concentrations of MPP^+^ at 10 and 200 μM can enhance autophagosome in an earlier stage of autophagy but impair autophagosome degradation, leading to cell death. Due to the assembly of MPP^+^ exposure, alpha-synuclein is then upregulated in SH-SY5Y cells [35,36]. Interestingly, Andro pretreatment resulted in decreased p62 expression, leading to decreasing alpha-synuclein accumulation in the cell, as shown in Figure 2C. On the contrary, treatment with 3-MA showed a decline in autophagy induction. Therefore, Andro provides neuroprotection by enhancing mitophagy and autophagy, which eliminates damaged mitochondria and alpha-synuclein accumulations.

As a consequence of defective mitochondria accumulation, ROS generation is increased, which impairs energy production. In addition, the overproduction of ROS-mediated oxidative stress can promote the release of cytochrome c, which subsequently triggers apoptosis of dopaminergic neurons [37]. Recently, it was reported that MPP^+^ induced intracellular ROS generation in cerebellar granule neurons, neuroblastoma, and N27 dopaminergic cells from damaged mitochondria, which caused apoptotic cell death [38,39]. We also found that Andro reduced ROS generation produced by MPP^+^ in SH-SY5Y cells by inducing Nrf2 antioxidant activation. Likewise, Andro exhibited an antioxidant effect against oxidative stress in the AD model based on the activation of the Nrf2 pathway [40]. Several studies reported that Nrf2 is a crucial defensive mechanism against oxidative stress through its ability to upregulate antioxidant and detoxifying genes in age-related diseases, including AD and PD [16,41]. Normally, Nrf2 is localized in the cytosol and interacts with KEAP1 under a basal condition. This interaction between KEAP1 and Nrf2 promotes Nrf2 ubiquitination that subsequently leads to proteasomal degradation [42,43]. Oxidative stress and Nrf2 activators can trigger Nrf2 activation by releasing Nrf2 from KEAP1 and allowing Nrf2 to translocate into the nucleus, where it activates the expressions of antioxidant enzymes to eliminate ROS and facilitate neuronal survival in many neurological diseases [44,45,46,47]. In our present study, we found that MPP^+^ decreased Nrf2 nuclear translocation while pretreatment with Andro facilitated Nrf2 nuclear translocation and increased expressions of SOD1, CAT, HO-1, and GSTP1 genes expressions, as well as SOD1 and GPx enzyme activities. These findings indicated that Andro decreased oxidative stress and protected cells from apoptosis by promoting antioxidant activities through the activation of the Nrf2 pathway. Andro might also directly promote cell survival as it was found that pretreatment with Andro diminished pro-apoptotic proteins, including caspase-3, BAX, and cytochrome c, while increased Bcl2, an anti-apoptotic protein. Accordingly, Andro directly promoted the expression of TH, which was responsible for DA synthesis in SH-SY5Y cells.

Taken together, our study demonstrated that Andro showed a neuroprotective effect against MPP^+^-induced neurotoxicity by mediating mitophagy and autophagy, which facilitated the elimination of defective mitochondria and alpha-synuclein degradation. Concomitantly, Andro activated the antioxidant Nrf2 pathway to counteract ROS build-up from defective mitochondria, which directly attenuated cell apoptosis. Thus, it is to be hoped that this study provides evidence to support the neuroprotective ability of Andro that may be considered as a potential therapeutic or supplement in the prevention of PD. However, further studies are needed to determine the proposed properties in in vivo mouse models, as well as stringent clinical trials in humans.

## 4. Materials and Methods

### 4.1. Cell Culture

Neuroblastoma SH-SY5Y cell line (ATCC# CRL-2266) and HFF cell line (ATCC# SCRC-1041) were purchased from American Type Culture Collections (ATCC, Manassas, Virginia, USA). SH-SY5Y and HFF cell lines were cultured in Dulbecco’s Modified Eagle’s Medium/Nutrient Mixture F-12 (DMEM/F-12) and DMEM high glucose (4.5 g/L), respectively, and supplemented with 10% fetal bovine serum (FBS) and 1% penicillin/streptomycin at 37 °C in a relatively humidified atmosphere with 5% CO_2_. The cultured cells were maintained at 37 °C in 5% CO_2_. The media were changed 2–3 times per week. SH-SY5Y and HFF cells have been used between passage numbers 31–34 and 17–20, respectively.

### 4.2. Cell Viability Assay

Andrographolide was purchased from Sigma-Aldrich (St. Louis, MO, USA, 365645). The purity (TLC) of this compound is ≥ 98%, with a formula weight of 350.45 g/mol. To examine the ability of andrographolide to prevent SH-SY5Y cell death from MPP^+^ exposure, SH-SY5Y cells (8000 cells/well) were cultured in 96-well plates for 24 h in a humidified 5% CO_2_ atmosphere at 37 °C. Cells were incubated with Andro that was dissolved in dimethyl sulfoxide (DMSO for cell culture; Sigma-Aldrich, USA) at designated concentrations for 24 h. After that, the solution was removed and treated with 1.5 mM of MPP^+^ for 24 h. Ultimately, cell viability was determined by adding 5 mg/mL MTT (3-(4,5-dimethylthiazol-2-yl)-2,5-diphenyltetrazolium bromide; Sigma-Aldrich, USA) for 2 h. At the indicated time, the supernatant was discarded, and formazan crystals were dissolved by adding 100 µL of dimethyl sulfoxide (DMSO; Sigma-Aldrich, USA). The absorbance was read by a microplate reader (Versamax Spectrophotometer, MA, USA) at 570 nm and 690 nm as modified from [48]. The percentage of cell viability was calculated in comparison with the untreated group.

### 4.3. Mitochondrial Membrane Potential Analysis

Mitochondrial membrane potential (MMP) is an important indicator of mitochondrial function. JC-1 Mitochondrial membrane potential assay kit (Abcam, ab113850, Cambridge, UK) was used to determine MMP. JC-1 staining indicates the healthy mitochondria by emitting red fluorescence, whereas unhealthy cells showed a decrease in red fluorescence and an increase in green fluorescence, which represent the remaining monomers in the cytoplasm. Hence, the ratio of red/green serves as an indicator of loss of MMP. The experiment was conducted as previously reported [49]. Cells were seeded at 15,000 cells/well in a black and flat bottom 96-well microplate and incubated with Andro at designated concentrations for 24 h, followed by treatment with MPP^+^ at 1.5 mM for 16 h. Then, the cells were washed with PBS and incubated with 10 µM of JC-1 dye in a dark room at 37 °C and 5% CO_2_ for 10 min. After washing with PBS, the quantification of relative fluorescence between red and green intensity was measured by a fluorescence microplate reader (TECAN Spark 10M, Bioexpress, Männedorf, CH, USA). The red fluorescence intensity (polymerized form of JC-1) was measured at an Ex/Em of 535/590 nm, and the green fluorescence intensity (monomerized form of JC-1) was measured at an Ex/Em of 490/530 nm. Furthermore, JC-1-stained cells were observed under a fluorescence microscope (Olympus BX53, Tokyo, Japan).

### 4.4. Immunofluorescence Staining

SH-SY5Y cells were grown on poly-l-lysine coated coverslips to improve cell attachment in 6-well culture plates at a density of 1.5 × 10^5^ cells per well. Then, cells were pretreated with Andro at designated concentrations for 24 h, followed by exposure to 1.5 mM MPP^+^ for 16 h. After that, treated cells were incubated with MitoTracker Red (Cell Signaling) for 30 min at 37 °C, then fixed with cold methanol for 15 min at −20 °C and washed three times with PBS for 5 min. Treated cells were permeabilized with 0.25% Triton-X-100 in PBS and blocked with 1% bovine serum albumin, 10% normal goat serum, and 0.3% glycine in PBST for 2 h. Subsequently, cells were incubated with anti-LC3 primary antibody (Cell Signaling Technology, Danvers, MA, USA) at 1:200 in a dark humidity box at 4 °C overnight, then incubated with a secondary antibody (Invitrogen, Alexa Fluor 488) goat anti-rabbit IgG (H+L) at 1:500 dilution for 2 h. Finally, the images were taken with a fluorescence microscope (Olympus BX53, Tokyo, Japan).

### 4.5. Intracellular ROS Measurement

To determine the ROS generation induced by MPP^+^ in SH-SY5Y cells, the ROS level was examined by staining with Dichloro-dihydro-fluorescein diacetate (DCFH-DA), a fluorescence probe used for detecting the level of ROS. Intracellular ROS was measured by a protocol modified from a previous report [50]. SH-SY5Y cells were cultured in a black and flat bottom 96-well microplate and exposed to designated concentrations of Andro for 24 h, followed by treatment with 1.5 mM MPP^+^ for 2 h. After washing with cold PBS, cells were incubated with 5 µM DCFH-DA (ThermoFisher Scientific, Waltham, MA, USA) in phenol red-free culture medium for 45 min at 37 °C in a dark room. Finally, the fluorescence intensity was detected at excitation and emission wavelengths of 488 nm and 525 nm by TECAN Spark 10M (Bioexpress, Männedorf, CH). Alternatively, SH-SY5Y cells were cultured in a 12-well microplate and exposed to designated concentrations as mentioned above, then observed under a fluorescence microscope (live-cell fluorescence imaging system, IX-83ZDC).

### 4.6. Nrf2 Nuclear Translocation

Nuclear translocation of Nrf2 was evaluated by immunofluorescence staining following a protocol previously reported [15]. SH-SY5Y cells were plated on the coverslips. After being treated with designated concentrations of Andro for 24 h and MPP^+^ for another 16 h, cells were washed three times with PBS and fixed with 4% paraformaldehyde for 15 min at room temperature. Treated cells were then washed three times with PBS and blocked with a blocking buffer (containing 0.3% Triton X-100 and 1% BSA in PBS) for 1 h. Afterward, cells were incubated with anti-Nrf2 antibody (Abcam, ab137550) at 1:200 dilution overnight at 4 °C, followed by a secondary antibody goat anti-rabbit IgG (H+L) (Invitrogen, Alexa Fluor 488) at 1:500 dilution for 1 h at 37 °C. The nuclei were counterstained with DAPI. The fluorescence images were taken by fluorescence microscope (Olympus BX53, Tokyo, Japan), and the intensity of Nrf2 expression was quantified by Image J software version 1.52k.

### 4.7. Real-Time Quantitative Reverse Transcription Polymerase Chain Reaction (RT-qPCR) for Determinations of Nrf2-Dependent Antioxidant Genes

SH-SY5Y cells were treated with or without 2 mM of NAC or pretreated with 1.5 µM of Andro for 12 h, followed by incubation with or without 1.5 mM of MPP^+^ for 8 h. After treatment, RNA was extracted by Total RNA Mini Kit (Blood/Culture cell) (Geneaid, RB300, New Taipei, Taiwan). Then, iScript Reverse Transcription Supermix (Bio-Rad, 170-8841, Heracles, CA, USA) was used for reverse RNA to cDNA. Real-time qPCR was performed by using 10 µL iTaq Universal SYBR Green Supermix (Bio-Rad, 172-5121), and PCR conditions were optimized by using a real-time thermocycler (Bio-rad/CFX96 touch). Nrf2-dependent antioxidant genes were amplified by using the following primers: GAPDH (forward: 5′-GACAGTCAGCCGCATCTTCT-3′, reverse: 5′-GCGCCCAATACGACCAAATC-3′); SOD1 (forward: 5′-GATGACTTGGGCAAAGGTGG-3′, reverse: 5′-TACACCACAAGCCAAACGACT-3′); CAT (forward: 5′-CTTCGACCCAAGCAACATGC-3′, reverse: 5′-GCGGTGAGTGTCAGGATAGG-3′); HO-1 (forward: 5′-AGGGAATTCTCTTGGCTGGC-3′, reverse: 5′-GACAGCTGCCACATTAGGGT-3′); and GSTP1 (forward: 5′-AAGTTCCAGGACGGAGACCT-3′, reverse: 5′-AAGTTCCAGGACGGAGACCT-3’). Fold change in gene expression was calculated by using 2^−ΔΔCt^ method.

### 4.8. Western Blot Analysis

SH-SY5Y cells were cultured on a 6-well plate in culture medium for 24 h. Cells were treated with or without an autophagy inhibitor, 3-MA at 2 mM (Sigma-Aldrich, 5142-23-4) for 2 h, or treated with NAC at 2 mM, or incubated with designated concentrations of Andro for 24 h followed by exposure with MPP^+^ for 16 h. Western blot was performed as previously described [18,19]. Treated cells were lysed with 1xRIPA buffer (25 mM Tris HCl pH 7.6, 150 mM NaCl, 1% NP-40, 1% sodium deoxycholate, 0.1% SDS) and PMSF as a non-specific protein inhibitor on ice for 10 min. To investigate the nuclear translocation of Nrf2, nuclear protein extraction was performed by using Nuclear Extraction Kit (Abcam, ab113474). These lysates were centrifugated at 12,000 RPM for 15 min at 4 °C. The supernatants were collected, and protein concentrations were estimated by BCA protein assay kit (ThermoFisher Scientific, 23225). Twenty micrograms of protein were electrophoresed and transferred to nitrocellulose membranes. These nitrocellulose membranes were blocked with TBST containing 5% BSA for 2 h and then incubated overnight at 4 °C with primary antibodies directed against caspase-3 (Cell Signaling Technology, #9662) at 1:1000, cytochrome c (Abcam, ab110325) at 1:1000, Bax (Cell Signaling Technology, #2774) at 1:700, Bcl-2 (Cell Signaling Technology, #4223) at 1:700, TH (Abcam, ab112) at 1:500, alpha-syn (Cell Signaling Technology, #2647) at 1:700, β-actin (Abcam, ab8227) at 1:1000, mTOR (Cell Signaling Technology, #2972) at 1:1000, p- mTOR (Cell Signaling Technology, #2971) at 1:1000, Beclin1 (Cell Signaling Technology, #3738) at 1:1000, LC3A/B (Cell Signaling Technology, #4108) at 1:700, LAMP1 (Cell Signaling Technology, #9091) at 1:1000, SQSTM1/p62 (Cell Signaling Technology, #5114) at 1:1000, PINK 1 (Abcam, ab216144) at 1:1000, Parkin (Abcam, ab77924) at 1:700, Nrf2 (Abcam, ab137550) at 1:700, Lamin B1 (Abcam, ab65986) at 1:700, KEAP1 (Cell Signaling Technology, #4678) at 1:700, and HO-1 (Abcam, ab13243) at 1:1000 followed by HRP-conjugated anti-rabbit or anti-mouse IgG (Abcam, ab6721 and ab mouse ab6789) at 1:700 at 1:5000 dilution with TBST at room temperature for 2 h. Chemiluminescent signal was observed by NovexTM ECL chemiluminescent substrate reagent kit (ThermoFisher Scientific, MA, USA) on chemiluminescent gel document (Alliance Q9 mini, VT, USA) and was analyzed by Image J software.

### 4.9. Antioxidant Enzymes Detection Assay

To measure the activities of antioxidant enzymes (SOD and GPx), the cells were treated with designated concentrations of Andro for 24 h, followed by MPP^+^ for 16 h, then lysed, and the supernatants were collected. Then, the enzyme activities were determined by using SOD and GPx Activity Assay Kit (Abcam, ab65354, and ab102530). The absorbance was evaluated by a microplate reader (Versamax Spectrophotometer, USA) at 450 nm and 340 nm for SOD and GPx activities, respectively.

### 4.10. Statistical Analysis

All values are presented as the mean ± standard deviation (SD). Data were collected from at least three independent experiments and analyzed using GraphPad Prism software version 8.2.1 (GraphPad Software, San Diego, CA, USA). Comparisons among multiple groups were performed using one-way ANOVA with Dunnett’s multiple comparisons test. One sample t-test was used to compare with the control group without SD. A *p*-value < 0.05 was considered statistically significant.

## 5. Conclusions

The neuroprotective effect of andrographolide on a PD cellular model was demonstrated by using MPP^+^-induced neuronal toxicity in SH-SY5Y cells, in which andrographolide promoted mitophagy mediated by PINK/Parkin signaling pathway and autophagy as indicated by the upregulations of Beclin1, LC3, and LAMP1. These actions help to eliminate defective mitochondria and alpha-synuclein. Simultaneously, Andro promoted Nrf2 activation, which induced expressions of genes encoding antioxidant enzymes and enzyme activities to counteract oxidative stress. Andrographolide might also directly prevent cell apoptosis by upregulation of mitochondrial membrane potential and Bcl2 and downregulations of cytochrome c, BAX, and cleaved-caspase-3.

## Figures and Tables

**Figure 1 ijms-24-08528-f001:**
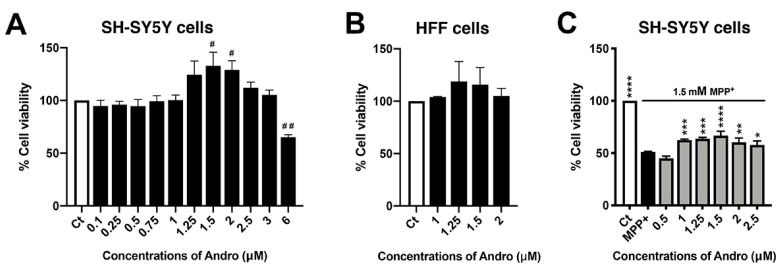
Cytotoxicity and protective effect of andrographolide (Andro) on SH-SY5Y and HFF cells. Cell viability was evaluated by MTT assay. (**A**) SH-SY5Y cells were treated with 0.1, 0.25, 0.5, 0.75, 1, 1.25, 1.5, 2, 2.5, 3, and 6 µM of Andro for 24 h. (**B**) HFF cells were treated with 1, 1.25, 1.5, and 2 µM of Andro for 24 h. (**C**) Neuroprotective effect of Andro on MPP^+^-induced SH-SY5Y cells toxicity. SH-SY5Y cells were treated with or without 0.5, 1, 1.25, 1.5, 2, 2.5 µM of Andro for 24 h, followed by treatment with MPP^+^ 1.5 mM for 24 h. * *p* < 0.05, ** *p* < 0.01, *** *p* < 0.001, **** *p* < 0.0001—statistical significance versus MPP^+^-treated group. # *p* < 0.05, ## *p* < 0.01—statistical significance versus control group (data are mean ± SD, *n* = 3).

**Figure 2 ijms-24-08528-f002:**
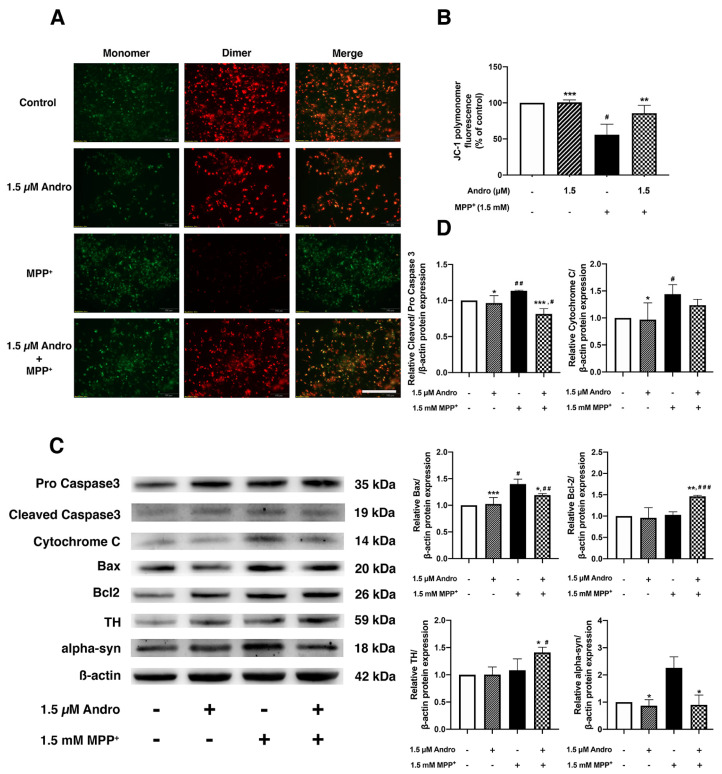
Protective effect of andrographolide on MPP^+^-induced neuronal death through inhibition of apoptotic pathway. SH-SY5Y cells were treated with or without 1.5 µM of Andro for 24 h, followed by exposure with MPP^+^ 1.5 mM for 16 h. (**A**) MMP was assessed by JC-1 staining. Green JC-1 monomers indicate depolarized MMP and red JC-1 dimers indicate normal MMP. Merged panel represents colocalization of green monomer and red dimer fluorescences. Scale bar: 50 µm. (**B**) Quantification of relative fluorescence between red and green in each group was measured by a microplate reader (*n* = 3). (**C**) Representative immunoblots of pro-caspase-3, cleaved-caspase-3, cytochrome c, Bax, Bcl2, TH, and alpha-synuclein. (**D**) Quantification of relative protein bands density normalized by β-actin in SH-SY5Y cells treated with or without 1.5 µM of Andro for 24 h, then treated with 1.5 mM of MPP^+^ for 16 h. * *p* < 0.05, ** *p* < 0.01, *** *p* < 0.001-statistical significance versus MPP^+^-treated group. # *p* < 0.05, ## *p* < 0.01, ### *p* < 0.001-statistical significance versus control group (data are mean ± SD, *n* = 3).

**Figure 3 ijms-24-08528-f003:**
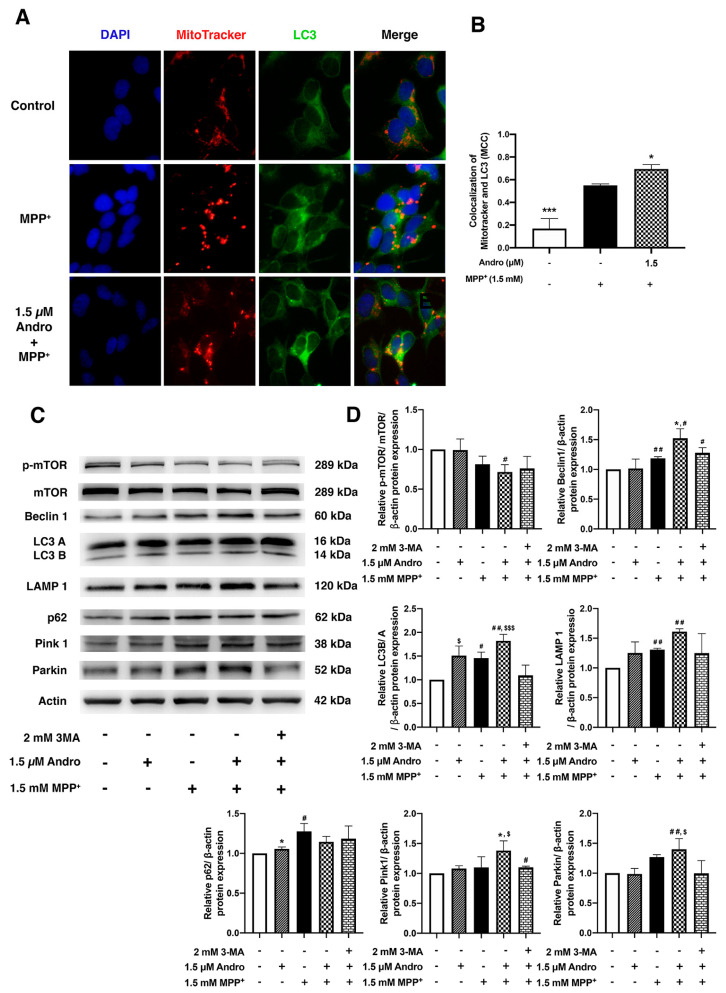
Protective effect of andrographolide against MPP^+^-induced SH-SY5Y cell death by enhancing autophagy and mitophagy induction. SH-SY5Y cells were incubated with or without 1.5 µM of Andro for 24 h, followed by incubation of MPP^+^ 1.5 mM for 16 h. (**A**) Representative immunofluorescence images of MitoTracker Red and LC3 (green) and nuclei were counterstained with DAPI (blue). Merged panel represents colocalization of mitophagy induction. Scale bar: 50 µm. (**B**) The Manders’ coefficient (MCC) was calculated from 3 independent experiments (at least 6 random fields were analyzed per condition) and quantified by using ImageJ. (**C**) Immunoblot of mTOR, p-mTOR, Beclin1, LC3B/A, LAMP1, p62, PINK1, Parkin. (**D**) Quantification of bands intensity normalized by β-actin in SH-SY5Y cells treated with or without 2 mM of 3-MA prior to incubation with or without 1.5 µM of Andro for 24 h, then treated with 1.5 mM of MPP^+^ for 16 h. * *p* < 0.05, *** *p* < 0.001-statistical significance versus MPP^+^-treated group. # *p* < 0.05, ## *p* < 0.01-statistical significance versus control group. $ *p* < 0.05, $$$ *p* < 0.001-statistical significance versus 3-MA group (data are mean ± SD, *n* = 3).

**Figure 4 ijms-24-08528-f004:**
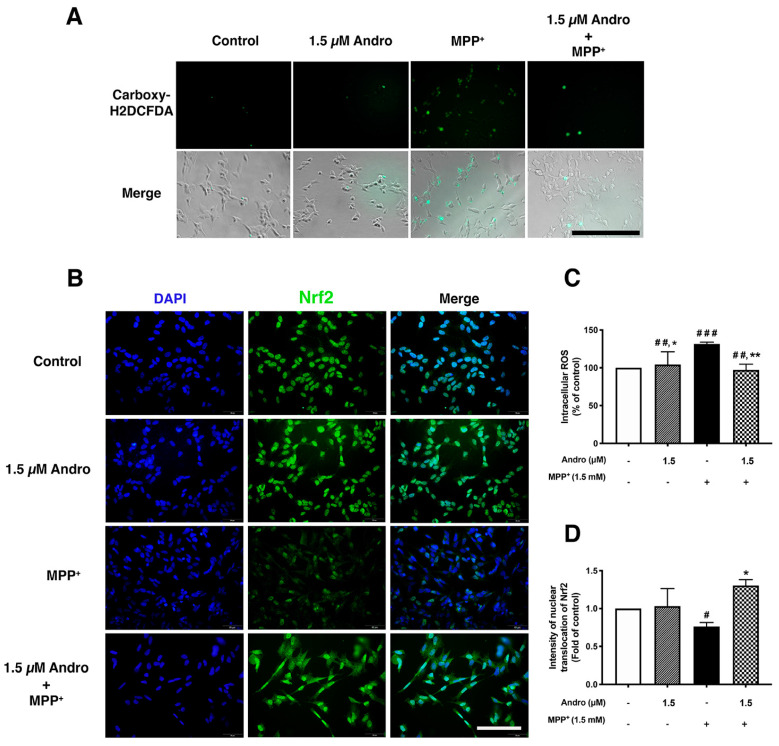
Antioxidant effect of andrographolide via Nrf2 activation in MPP^+^-induced toxicity in SH-SY5Y cells. The ROS generation was induced by incubation of SH-SY5Y cells with MPP^+^ 1.5 mM for 2 h after incubation with or without 1.5 µM of Andro for 24 h. (**A**) Intracellular ROS was measured by carboxy-DH2DCFDA staining, and green fluorescence was observed under IX83 Inverted microscope. Scale bar: 100 µm. (**B**) Representative images of Nrf2 staining (green) and DAPI (blue) which indicated nuclear translocation by observing the colocalization of Nrf2 and DAPI in SH-SY5Y cells treated with 1.5 µM of Andro for 24 h prior to being treated with 1.5 mM of MPP^+^ for 16 h. Scale bar: 50 µm. (**C**) Quantification of intracellular ROS fluorescence intensity in each group was measured by microplate reader. (*n* = 3). (**D**) Quantification of Nrf2 nuclear translocation intensity was calculated from 3 independent experiments (at least 6 random fields were analyzed per condition) and quantified by using ImageJ. * *p* < 0.05, ** *p* < 0.01-statistical significance versus MPP^+^-treated group. # *p* < 0.05, ## *p* < 0.01, ### *p* < 0.001-statistical significance versus control group (data are mean ± SD, *n* = 3).

**Figure 5 ijms-24-08528-f005:**
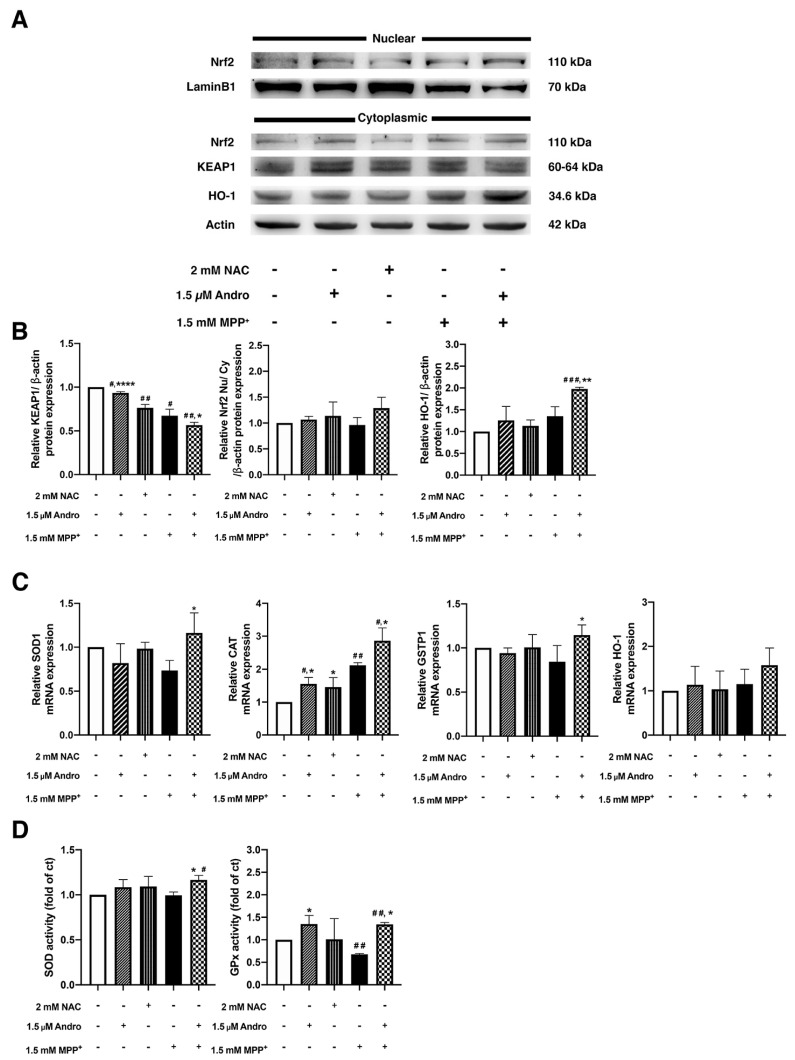
Antioxidant effect of andrographolide via Nrf2 activation in MPP^+^-induced SH-SY5Y cells. (**A**) The immunoblots of Nrf2 in nuclear and cytoplasmic fractions, KEAP1 and HO-1. (**B**) Quantification of band intensity normalized by β-actin. (**C**) The antioxidant mRNA levels of SOD1, CAT, GSTP1, and HO-1 were evaluated by RT-qPCR assay. SH-SY5Y cells were incubated with or without 1.5 µM of Andro or 2 mM NAC for 12 h, followed by incubation of MPP^+^ 1.5 mM for 8 h. (**D**) The antioxidant activities of SOD and GPx enzymes in SH-SY5Y cells treated with or without 1.5 µM of Andro or 2 mM of NAC for 24 h prior to incubated with 1.5 mM of MPP^+^ for 16 h. * *p* < 0.05, ** *p* < 0.01, **** *p* < 0.0001-statistical significance versus MPP^+^-treated group. # *p* < 0.05, ## *p* < 0.01, ### *p* < 0.001-statistical significance versus control group (data are mean *±* SD, *n* = 3).

## Data Availability

The data supporting the conclusion in this study are available on request from the corresponding author.

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
