# Peer review of "Neuroprotection of Andrographolide against Neurotoxin MPP+-Induced Apoptosis in SH-SY5Y Cells via Activating Mitophagy, Autophagy, and Antioxidant Activities"

_ijms, 2023, doi:10.3390/ijms24108528_

Round 1

Reviewer 1 Report

The neuroprotective effect of Andro against MPP+-induced neurotoxicity by mediating mitophagy and autophagy has been established in the work. Using various independent approaches, the role of Andro in the activation of the antioxidant Nrf2 pathway has been convincingly proven.

The work was done qualitatively, the results are consistent and well presented. In general, the work makes a good impression. There are no significant comments, but there are a number of shortcomings:

1. It is necessary to give a decoding of the abbreviation MPP at the first mention, that is, in the abstract section.

2. It is not clear where the authors took the drug andrographolide (Andro). It is necessary to write in more detail about the known mechanisms of its action in the introduction section and provide information about this drug in the materials and methods section.

Author Response

Our response to reviewer's comments is in the attached file. 

Reviewer 2 Report

In the article entitled "Neuroprotection of Andrographolide against neurotoxin MPP+-induced apoptosis in SH-SY5Y cells via activating mitophagy, autophagy, and antioxidant activities" the authors demonstrate with several experiments in SH-SY5Y cell culture the neuroprotective effect of Andrographolide. The study suggests that Andrographolide having neuroprotective effects potentiated mitophagy and alpha-synuclein clearance through autophagy, as well as increased antioxidative capacity.

The study provides evidence that Andrographolide could be considered as a potent supplement in the prevention of Parkinson's disease.

Of course, it is important to do in vivo studies, as well as rigorous human clinical trials, to determine the optimal dose and duration of treatment.

It is a complete in vitro study, so this article is a suitable  to be accepted in the Journal.

Author Response

(The authors gave the same response as above.)

Reviewer 3 Report

The Ms. by Prasertsuksri et al. confirms the neuroprotective capacity of andrographolide (Andro) in an in vitro system for Parkinson's dissease, which was previously demonstrated by Geng et al. (Br J Pharmacol 2019;176:4574-4591) in vivo. It is important to note that this article has not been referred.

Additional comments:

1.      The text should be revised by a native English speaker.

2.      For clarity, the authors should explain the rational of the experiments that are described in the Results section. For instance, what is the rational for red vs green fluorescence in Fig. 2A,B (lines 115-117)? Lines 178-180: explain why LC3A/B, LAMP1 and Beclin are chosen.

3.      Fig. 1B should include cell viability for 2.5, 3 and 6 uM Andro since the addition of 2 uM Andro to HFF cells results in similar viability levels as control conditions while it improves cell viability in SH-SY5Y cells. Could it be that Andro potentiates cell death at higher concentrations in HFF cells as compared to SH-SY5Y cells?

4.      Supplementary Figures 1 and 2 are missing.

There are multiple examples of text that is difficult to understand. Just as a few examples:

- Line 13: Andro has been shown several pharmacological actions (instead of "Andro has been shown to induce several pharmacological actions").

- Line 158: we observed the mitophagy (instead of "we studied the mitophagic effect").

Author Response

(The authors gave the same response as above.)
